# Prospect on Rare Earth Elements and Metals Fingerprint for the Geographical Discrimination of Commercial Spanish Wines

**DOI:** 10.3390/molecules25235602

**Published:** 2020-11-28

**Authors:** Claudia Cerutti, Raquel Sánchez, Carlos Sánchez, Francisco Ardini, Marco Grotti, José-Luis Todolí

**Affiliations:** 1Department of Chemistry and Industrial Chemistry, University of Genoa, Via Dodecaneso 31, 16146 Genoa, Italy; claudia.cerutti1993@gmail.com (C.C.); ardini@chimica.unige.it (F.A.); grotti@unige.it (M.G.); 2Department of Analytical Chemistry, Nutrition and Food Science, University of Alicante, P.O. Box 99, 03080 Alicante, Spain; carlos.sr.ua@gmail.com (C.S.); jose.todoli@ua.es (J.-L.T.)

**Keywords:** ICP-MS, rare earth elements (REEs), wine, metals, designation of origin (PDO) discrimination

## Abstract

This paper presents a novel tool for Spanish commercial wine discrimination according to their designation of origin (PDO). A total of 65 commercial wines from different Spanish designation of origin (Alicante, Bullas, Campo de Borja, Jumilla, Castilla la Mancha, Ribeiro, Ribera de Duero, Rioja, Rueda, Utiel-Requena, Valdepeñas and Valencia) were characterized. The rare earth elements (REEs) content was determined by a high-temperature torch integrated sample introduction system (hTISIS) coupled to inductively coupled plasma mass spectrometry (ICP-MS). The REE content was used to draw characteristic PDOs radar charts. Results indicated that the REEs fingerprint provides a good prospect to discriminate the different Spanish PDOs, except for Alicante, Castilla la Mancha, Jumilla, Utiel-Requena and Valdepeñas. Finally, for those PDOs that were not properly distinguished, a second fingerprint obtained from Ba, Co, Cr, Mn, Ni, Pb and V content was used for discrimination purposes.

## 1. Introduction

Wine is among the products in which geographical origin is considered as a quality mark. From a chemical point of view, wine could be considered as a complex mixture, containing water, ethyl alcohol and sugars, along with other inorganic and organic compounds [1,2,3]. Moreover, trace elements are present [4,5,6]. Endogenous metals come from the soil that vines are grown on and they are delivered to the wine through grapes. In addition, exogenous metals are associated with external impurities that may contaminate the wine during the growth of grapes or at different stages of the winemaking process, from harvesting to bottling and cellaring [4].

Wine elemental composition could provide relevant information on its quality, physico-chemical characteristics and geographical origin [4,5,6,7,8]. In fact, the concentration of elements strongly determines stability, organoleptic and nutrition characteristics of wine. Some metals, such as Cu, Fe, Mn, Ni and Zn mainly affect the organoleptic characteristics of wine (i.e., flavor, freshness, aroma, color and taste), due to the formation of precipitates (yeast, fining and filtration sediments) or clouding during wine fermentation, maturation and storage [4,8]. However, other elements, including As, Cd, Cu, Pb and Zn, are of great concern due to their toxicity [8]. As regards the link between the metal composition and the geographical origin, several publications concluded that the elemental profile can be a useful tool for provenance discrimination [5,6,9,10,11,12,13,14,15,16,17]. In particular, rare earth elements (REEs) have been used to distinguish wines from different regions of Italy [10,18], England [19], Spain [19], other regions of Europe [20,21], California [20,22], Australia [9,20] and South America [23]. REEs content is used as a tool for geographical discrimination because REEs composition at each step of winemaking process shows the original distribution in soil [18,24]. Moreover, modifications of REEs distribution are caused by additives used to promote fermentation [25].

The accurate determination of REEs and metals in wine samples is not a trivial task. In recent years, inductively coupled plasma mass spectrometry (ICP-MS) has been established as the analytical technique of choice for this purpose [4,9,10,11,23,26,27,28,29]. However, the introduction of organic samples into the plasma is still a challenge, as ICP techniques suffer from severe interferences caused by complex organic matrices, including matrix effects, plasma degradation and soot deposition at the injector tip and interface cones [30,31,32]. To circumvent them, several sample preparation approaches have been developed, such as sample dilution, conventional dry/wet sample digestion, microwave- or ultraviolet-assisted acid digestion, dealcoholization and analyte extraction. However, all these methods show some problems caused by the addition of reagents, potential sample contamination and degradation of limits of detection, among others [9,11,16,20,22,23,27,28,29,33,34,35,36,37].

As an alternative to these approaches, the use of the high temperature torch integrated sample introduction system (hTISIS) has been proposed and successfully applied for the routine analysis of wine samples [26]. The basic principle of this low sample consumption system relies on the achievement of complete aerosol evaporation before its introduction into the plasma source, thus accomplishing analyte transport efficiency close to 100% regardless of the sample matrix [26,38,39]. Additional advantages of the hTISIS over conventional sample introduction systems include the improvement of sensitivity and limits of detection, as well as the shortening of wash out times. However, an excessive amount of solvent reaching the plasma may degrade its thermal characteristics. Therefore, low sample flow rates (i.e., few tens of microliters per minute) must be selected.

Since the production of wines under a protected designation of origin (PDO) has a significant economic impact, there is a clear need of suitable tools to verify wine authenticity. The main goal of the present work was thus to investigate the possibility of identifying the Spanish geographical provenance of commercial wines by means of a REEs fingerprint combined with some complementary data of metal concentration. Hypothetically the REEs fingerprint could be used as an anti-fraud tool to classify wine according to its PDO, without applying statistical analysis. Moreover, REEs and metals content was determined by analyzing undiluted wine samples using the hTISIS coupled to ICP-MS.

## 2. Results

### 2.1. Quantification Method

A novel method for the elemental analysis of undiluted wine samples was optimized and validated in a previous work [26]. The method was based on the use of the hTISIS (working at 30 µL min^−1^ and 125 °C) coupled to ICP-MS and allowed the direct determination of trace elements in wine samples, with minimum sample manipulation and reduced analysis time. For REEs, the accuracy was tested by spiking experiments on two representative samples. Wine samples were spiked with a multi-elemental solution at a 50 µg kg^−1^ level. External calibration was performed with a set of standards containing 10% in ethanol and the ICP-MS intensities for non-spiked samples were subtracted to those of the corresponding spiked ones. Appendix A summarizes the recoveries found and for two representative wine samples and several REEs in both wine samples. It may be observed that for the selected operating conditions all the obtained data were included in between 100 ± 10%. However, the lowest recovery values were obtained for the white wine. In addition, the developed method proved to be adequate for routine wine analysis, as demonstrated by the analysis of ~70 samples during a 20-h long analytical session [26]. A quality control (QC) standard at the concentration of 10 µg kg^−1^ was analyzed every hour to assess a possible drift in the analytical signals, and the variation of the concentration for the QC samples throughout the analytical run was lower than 10%.

One of the most relevant analytical figures of merit in the method selection to quantify REEs and metal in wine samples is the detection limit (DL). This parameter is crucial for REEs determination since these elements are usually present at sub-µg kg^−1^ concentration level [21,23,28,40]. The method detection limits (DLs) were calculated according to the 3s_b_ criterion, where s*_b_* was the standard deviation of ten consecutive blank measurements. The method DLs achieved by hTISIS/ICP-MS were in the same order of those obtained in previous works (Table 1) [10,11,21,22,23,26,28,40], and proved to be adequate for REEs determination in wine samples. On the other side, DLs achieved by the hTISIS method for Ba, Co, Cr, Mn, Ni, Pb and V were significantly lower than those obtained by alternative methods, with improvement factors ranging from 2 to 40 [26]. This result was a combination of the direct injection of the sample and the increase in sensitivity with the use of a total sample consumption system, although these factors were partially balanced by the higher noise due to the high temperature of the hTISIS [26].

### 2.2. Metal Content in Wine Samples

Total metal content information in commercial wines is crucial as a quality assurance tool. Before obtaining the fingerprint of each analyzed wine, an in-depth evaluation of the metal content was carried out. In previous works it has been concluded that Ba, Cd, Co, Cr, Mn, Ni, Pb and V content could be used as a discrimination tool [4,6]. In the present work, Cd was not included for data analysis because it was only found in two Alicante samples.

The application of principal component analysis (PCA) yielded two principal components explaining almost 70% of the total variance. As it may be observed from the loading plot (Appendix A), all elements directly loaded on the first principal component, whereas the second one divided the analytes into two groups, which are positively (Mn, Ba, Pb, Co) or negatively (V, Ni, Cr) correlated to that component. Moreover, it can be noted in the score plot (Appendix A) that wines from Rueda could be distinguished from the other PDOs. Furthermore, wines from Alicante and La Mancha presented a widespread distribution along Component 1. Therefore, it was not possible to establish a direct correlation between the wine protected designations of origin and the Ba, Co, Cr, Mn, Ni, Pb and V contents. Additional criteria were, thus, needed to obtain a clearer discrimination among different geographical provenance of wines available in the Spanish market. The next step was, hence, to evaluate the concentration of REEs as a tool for wine identification.

### 2.3. REEs Content in Wine Samples

The REEs, except La and Lu, were at detectable levels in all the analyzed wine samples (Appendix A), whereas La and Lu were only found in some of the Rioja (Appendix A) and Alicante (Appendix A) samples. Therefore, these elements were not included in the data analysis to obtain the REEs fingerprint. REEs concentrations were comparable to those reported for wines from New Zealand [40] and Germany [21], and one order of magnitude higher than the concentrations found in wines from South America [23] and Italy [28]. These differences can be related not only to the geographical region, but also to the applied winemaking procedure, having an impact on the resulting elemental profile [4,21,22,25,41].

A general overview of the REEs content in wine samples was obtained by principal component analysis (Figure 1). For the PCA analysis only those PDOs that included at least five samples were considered (Appendix A). After removing the outlier RD4 sample and autoscaling the data, two principal components were built, collectively explaining 87% of the total variance. All REEs directly load on the first component (Figure 1a), meaning that their contents are higher in the samples on the right side of the score plot (Figure 1b). Moreover, the samples were also spread along the second component, with a positive load of a group of elements (Pr, Nd, Sm, Dy, Er and Yb), and a negative load of another group of elements (Eu, Gd, Tb, Ho and Tm). By the analysis of the PCA plots, it can be noted that the wines from Utiel-Requena (U1–U7) could be distinguished from the other PDOs, because of their lower REEs content. Furthermore, the wines from Valencia presented a widespread distribution, with extremely high REEs content in some samples (V2 and V3). Finally, the sixteen samples from the Rioja region were characterized by intermediate REEs concentrations and were grouped in the central part of the score plot. In fact, the variance for the Rioja samples was significantly lower than the total variance for all REEs (F-test, *p* < 0.05) except for Ho, Sm and Tb.

Variability in the samples considered in this work could be explained by the origin, the color, the grape variety and aging process. The REEs patterns in relation to the wine PDO will be discussed in Section 3.1. Concerning the color, some authors observed that, due to the differences in the production process, higher concentration of trace elements were expected in red wines than in white ones [42]. In fact, in the production of red wines, the grape-skin contact is longer. However, other studies did not highlight any difference in the concentration of trace elements [6]. Differences in the REEs content according to the color were evaluated by applying the Student’s t-test to Rioja and Alicante wines, for which both red and white wines samples were available (Appendix A). No significant differences were found, at the 95% significance level, except for Gd in Rioja wines, where the average concentration in red wines (3.3 µg kg^−1^) was significantly (*p* = 0.006) higher than that in the white ones (1.9 µg kg^−1^).

In order to explore the effect of grape variety, only monovarietal wines from the same region were taken into account, i.e., Tempranillo and Viura varieties from Rioja (Appendix A). No significant differences were found, either in the mean concentrations (t-test, *p* > 0.27) or in their variability (F-test, *p* > 0.23). The effect of the aging process was investigated by comparing the REEs content in wine samples RJ3 and RJ6, having the same color (red), region of origin (Rioja) and grape variety (Tempranillo and Garnacha), but subjected to different aging processes by the same producer (Appendix A). In particular, RJ3 was labeled “crianza”, since it was aged for 12 months in oak barrels and 24 months in bottle, whereas RJ6 was bottled without any aging process (“young”). It was observed that the REEs content was up to 1.4-times higher in the young wine than in the aged one for some elements, whereas for La and Gd concentrations were lower (Appendix A). In contrast, similar REEs concentrations were found for “reserve” (M1) and “young” (M4) wines from Castilla la Mancha (Appendix A), whereas their content was two times higher for M2 sample, labeled “crianza”, an intermediate aging period. Finally, the “crianza” wine from Ribera del Duero RD2 presented a lower concentration than the younger “roble” sample RD3 (Appendix A). Therefore, it was not possible to establish a clear connection between the aging process and REEs concentration.

## 3. Discussion

### 3.1. Classification of Wines with Different PDO According to Their REEs Content

REEs data were explored in terms of variability among samples belonging to the same PDO and among the different PDOs. Only the PDO in which the number of samples was equal or greater than five were considered (Appendix A). Moreover, outlier values were not included in the statistical evaluation. Fishers Least Significant Difference (LSD) test and ANOVA (Appendix A) showed that, for all the REEs elements, there were significant differences in the composition of wines having different PDOs, thus allowing to obtain specific REEs fingerprints. The F value was obtained by estimating the variation of the means between the different PDOs and dividing it by the estimation of the variation of the means within a given PDO. High F values were obtained when the variation in the mean concentration among the different PDOs was higher than the variation within a particular PDO, thus meaning that the means differed according to the PDO.

Rare earth elements fingerprints were obtained by using the “radar charts” shown in Figure 2. In each chart, the mean concentration value of each REEs was plotted as the distance from the center to the corresponding vertex and the line connecting all the points represents a specific REEs signature. Variability among samples of the same area (expressed as standard deviation) is also shown in the same plot as broken lines.

In this way, a characteristic profile for each PDO was obtained, except for Alicante (Figure 2a) and Castilla la Mancha (Figure 2b), which showed a quite similar REEs fingerprint, with the only difference of the Nd content. The geographical denomination of Castilla la Mancha covers areas of Albacete, Ciudad Real, Cuenca and Toledo (Appendix A). Four of the five samples included in this PDO were produced in Albacete, the closest area to the Alicante region (Appendix A). Thus, the small spatial separation of the regions might justify the limited differentiation. In contrast, characteristic REEs fingerprints were found for Valencia (Figure 2f) and Utiel-Requena (Figure 2g), despite their geographical proximity.

Radar plots were also built for the PDO in which the number of samples included in the study was lower than five (Figure 3). However, due to the low number of samples, the obtained profiles must be considered as preliminary ones. The Bullas (Figure 3a) and Jumilla (Figure 3b) areas are geographically close (Appendix A), but they show a different REEs fingerprint. However, the low number of representative samples caused overlaps between PDOs. Wines from Castilla la Mancha (Figure 2b), Valdepeñas (Figure 3c) and Jumilla (Figure 3b) presented a similar fingerprint. The former ones are part of the PDO of the autonomous region of Castilla la Mancha, whereas Jumilla is geographically close to that region (Appendix A). Finally, wines from Utiel-Requena (Figure 2g) and Campo de Borja (Figure 3d) presented a comparable REE-fingerprint in spite of their geographical distance.

### 3.2. Classification of Wines with Different PDO According to the Other Metals

For PDO in which a clear determination has not been established by using the REEs fingerprints, the possibility of distinguishing PDOs by alternative metals content (i.e., Ba, Co, Cr, Mn, Ni, Pb, V) has been evaluated. “Radar charts” have been obtained for those metals. Figure 4 shows the radar charts obtained for each PDO. Alicante and Castilla la Mancha presented a quite similar REEs fingerprint (Figure 2a,b). However, the new radar charts (Figure 4a,b) allowed the differentiation between both PDOs. In fact, Cr and Ni content were higher in Castilla la Mancha wines. Similar radar charts could be used to distinguish Castilla la Mancha (Figure 4b), Valdepeñas (Figure 4c) and Jumilla wines (Figure 4d). Utiel-Requena and Campo de Borja wines could also be distinguished by using Ba, Co, Cr, Mn, Ni, Pb and V radar charts (Figure 4e,f). The main difference between these PDOs was the Ba and V content.

## 4. Materials and Methods

### 4.1. Wine Samples

Sixty-five Spanish commercial wines were analyzed (Appendix A). Wine selection was based on consumption rates and covered twelve PDO (Alicante, Bullas, Campo de Borja, Jumilla, Castilla la Mancha, Ribeiro, Ribera de Duero, Rioja, Rueda, Utiel-Requena, Valdepeñas and Valencia) (Appendix A) of different vintage, color and grape variety. Red wines were elaborated with Cabernet Sauvignon, Garnacha, Merlot, Monastrell, Shiraz and Tempranillo grape varieties. The white grape varieties were: Chardonnay, Macabeo, Merseguere, Moscatel and Viura. Some of the wines were aged in barrels. Wine samples were filtered on 0.45-µm PTFE membranes (Filabet, Barcelona, Spain) and analyzed without any additional sample preparation.

### 4.2. Reagents and Materials

The quality control standard and calibration curve were prepared by using a 10 mg L^−1^ multi-element standard SCP33MS (SCP Science, QC, Canada) and 10 mg L^−1^ rare earth ICP-MS standard CMS-1 (Inorganic Ventures, Christiansburg, VA, USA) in 10% (*v*/*v*) ethanol using ultrapure water (Millipore, El Paso, TX, USA) and analytical-grade 96% ethanol (Panreac, Barcelona, Spain). Analyte concentrations ranged from 0.5 to 500 µg kg^−1^ for Ba, Cd, Co, Cr, Mn, Ni, Pb and V; and from 0.05 to 100 µg kg^−1^ for REEs. The internal standard stock solutions were prepared from 1000 mg L^−1^ Ge and Re standard solutions (HPS, Charleston, SC, USA) and 1000 mg L^−1^ Rh standard solution (SCP Science).

### 4.3. ICP-MS Instrumentation

An Agilent Technologies (Santa Clara, CA, USA) 7700x ICP-MS spectrometer, equipped with a high matrix introduction system (HMI) and the collision-reaction cell (CRC) operating in KED mode (He) was used. The main operating conditions are gathered in Appendix A. The hTISIS was equipped with a MicroMist nebulizer (Glass Expansion, Melbourne, Australia) connected to a 9-cm^3^ single-pass spray chamber heated by means of a copper coil [39]. Wine samples were delivered to the nebulizer at the flow rate of 30 µL min^−1^ by means of the Agilent G3160B autosampler, using 0.25-mm flared end PVC tubing (Glass Expansion, Melbourne, Australia).

### 4.4. Statistical Data Treatment

Variations between and within groups of samples were analyzed by one-way analysis of variance (ANOVA), whereas the statistically significant differences between mean values were evaluated by a Student’s t-test. Principal Component Analysis (PCA) was carried out using the open-source software R version 3.5.3 (Vienna, Austria), with the additional packages *Rcdmr* and *CAT* [43].

## 5. Conclusions

The application of hTISIS working at 30 µL min^−1^ and 125 °C in combination with ICP-MS allows the direct determination of REEs in wine samples, with minimum sample manipulation and high sample throughput, thus representing a suitable alternative for the routine analysis of wine samples.

Our data support the view that the REEs fingerprint can be used as a potential tool to discriminate the geographical origin of Spanish commercial wines. Although grape variety, color and winemaking procedures can modify the elemental profile, the obtained REEs fingerprints were able to distinguish among wines belonging to different PDOs (Figure 5). Moreover, for those PDOs in which it was not possible to establish a differentiation using the REEs fingerprint, a second fingerprint according to the content of Ba, Co, Cr, Mn, Ni, Pb and V was used to obtain a distinction among them (Figure 5). However, in order to definitively establish the REEs fingerprint as a robust anti-fraud tool, it will be necessary to enlarge the number of representative samples for each PDO, in order to be able to apply more robust classification and class-modeling techniques.

## Figures and Tables

**Figure 1 molecules-25-05602-f001:**
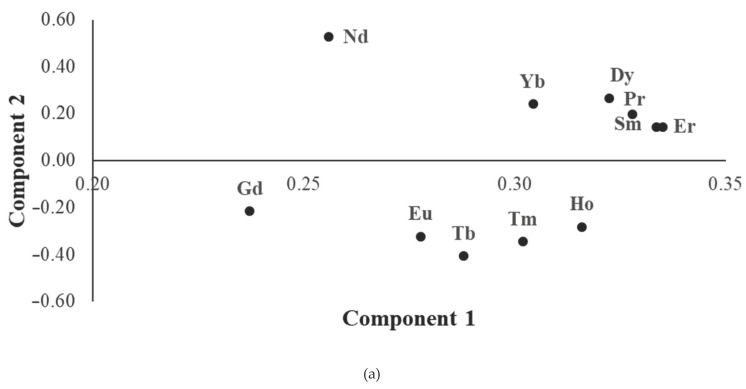
Principal components analysis performed on data expressed as concentration of REEs (Dy, Er, Eu, Gd, Ho, Nd, Pr, Sm, Tb, Tm and Yb) in wines from different designations of origin. (**a**) Loading plot; (**b**) score plot.

**Figure 2 molecules-25-05602-f002:**
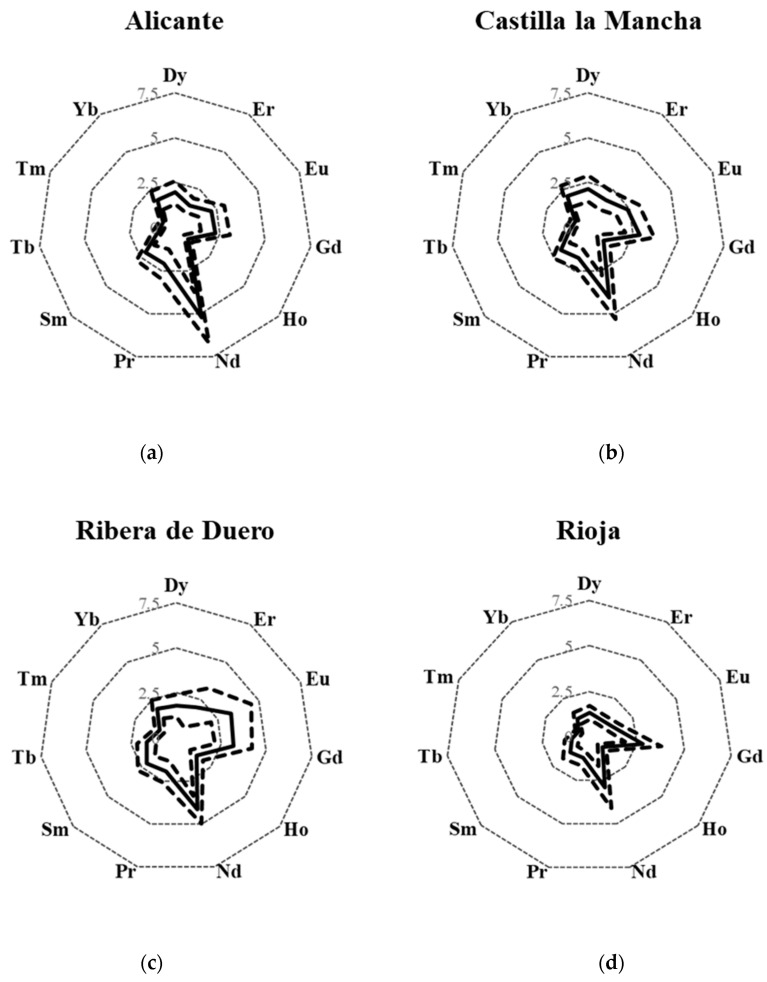
REEs (Dy, Er, Eu, Gd, Ho, Nd, Pr, Sm, Tb, Tm and Yb) fingerprint obtained for designation of origin. Continuous line: average; broken line: average ± standard deviation. Concentration units: µg kg^−1^. (**a**) Alicante; (**b**) Castilla la Mancha; (**c**) Ribera de Duero; (**d**) Rioja, (**e**) Rueda, (**f**) Valencia; and (**g**) Utiel-Requena.

**Figure 3 molecules-25-05602-f003:**
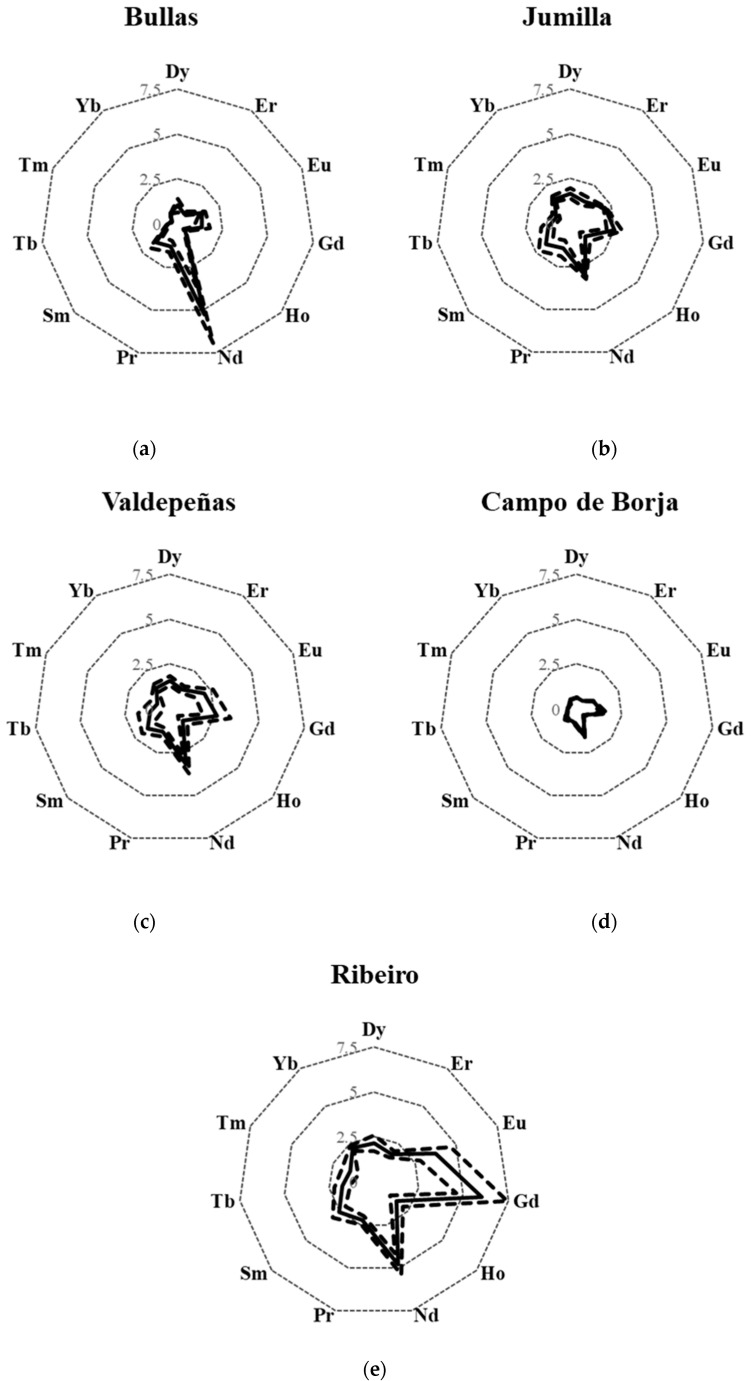
REEs (Dy, Er, Eu, Gd, Ho, Nd, Pr, Sm, Tb, Tm and Yb) fingerprint obtained for each designation of origin. Continuous line: average; broken line: average ± standard deviation. Concentration units: µg kg^−1^. (**a**) Bullas; (**b**) Jumilla; (**c**) Valdepeñas; (**d**) Campo de Borja, and (**e**) Ribeiro.

**Figure 4 molecules-25-05602-f004:**
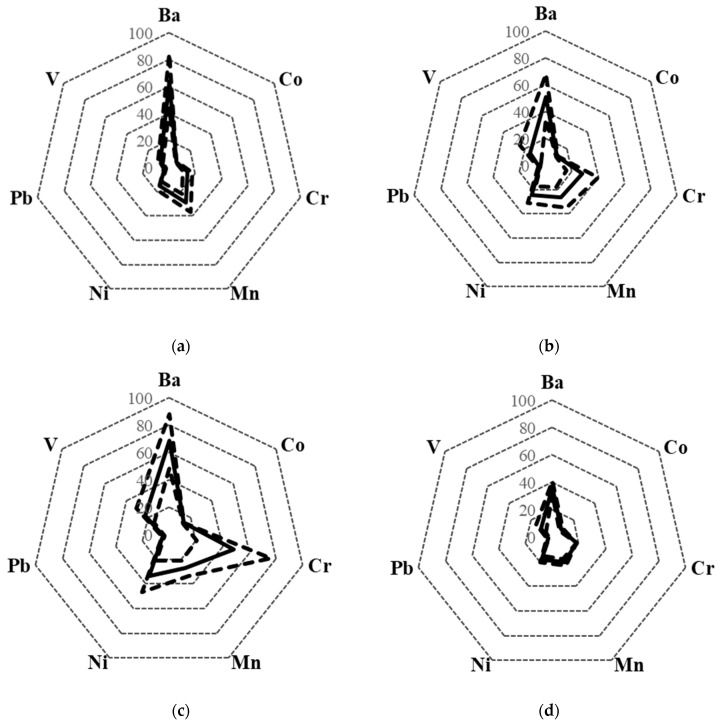
Alternative metal (Ba, Co, Cr, Mn, Ni, Pb and V) fingerprint obtained for (**a**) Alicante; (**b**) Castilla la Mancha; (**c**) Valdepeñas; (**d**) Jumilla; (**e**) Utiel-Requena; (**f**) Campo de Borja. Continuous line: average; broken line: average ± standard deviation. Concentration units: µg kg^−1^. Ba concentration divided by 2 and Mn concentration divided by 25.

**Figure 5 molecules-25-05602-f005:**
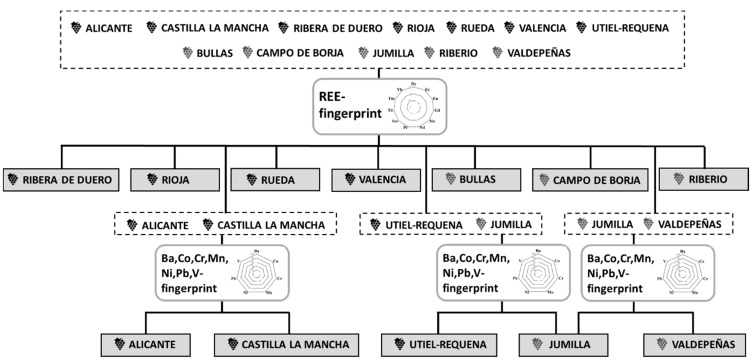
Scheme of separation of the different protected designation of origin (PDOs) based on the REEs (Dy, Er, Eu, Gd, Ho, Nd, Pr, Sm, Tb, Tm and Yb) and the alternative metals (Ba, Co, Cr, Mn, Ni, Pb and V) fingerprints.

**Table 1 molecules-25-05602-t001:** Method detection limits of various inductively coupled plasma mass spectrometry (ICP-MS)-based analytical methods for rare earth elements (REEs) determination in wine samples (values in ng kg^−1^).

Element	hTISIS/ICP-MS	Ref. [10]	Ref. [21]	Ref. [22]	Ref. [23]	Ref. [28]	Ref. [28]
*^139^La*	6.2	3.0	1.8	0.6	6.6		
*^141^Pr*	0.4	1.0	0.9	0.3	1.3	0.5	3.0
*^146^Nd*	2.3	5.0	1.0	0.7	10.0	2.0	3.0
*^147^Sm*	1.6	4.0	1.3	2.0	9.4	2.0	2.0
*^153^Eu*	0.4	1.0	2.0	0.005	1.2	0.5	3.0
*^157^Gd*	1.9	3.0	0.7	1.0	2.7	1.0	4.0
*^159^Tb*	1.8	1.0	0.2		0.5		
*^163^Dy*	0.2	1.0	1.7	0.4	2.3	1.0	5.0
*^165^Ho*	0.1	1.0	0.4	0.6	1.0	0.4	3.0
*^166^Er*	0.7	1.0	1.2	0.8	2.3	1.0	3.0
*^169^Tm*	0.1	1.0	0.1	0.3	0.5	1.0	2.0
*^172^Yb*	0.6	1.0	0.9	1.0	1.3	1.0	3.0
*^175^Lu*	0.2	1.0	0.2		0.2		
Sample preparation	None	Acidification	Microwave digestion	3-fold dilution	10-fold dilution	3-fold dilution	Microwave digestion

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
