# Peer review of "Prospect on Rare Earth Elements and Metals Fingerprint for the Geographical Discrimination of Commercial Spanish Wines"

_molecules, 2020, doi:10.3390/molecules25235602_

Round 1

Reviewer 1 Report

The paper presents interesting results on the using of rare earth elements fingerprint for the geographical discrimination of Spanish wines.

The concept of using REE for discrimination of various wines is not novel, but the authors complemented it with analysis of other metals. Additionally, they applied a novel direct procedure for determination of elements in wines by ICP-MS.

Detailed remarks:

Introduction: lines 43-46: not all papers cited in this sentence describe using of REE for discrimination of wines. Please check and correct. Please  describe also why REE can be used for geographical discrimination.

Novelty of the paper is not highlighted. Please, improve it.

Different abbreviations are used in the paper for wine: PDO vs DPO. Please check and correct if necessary.

Section 2.1. The short description of results presented in Supplementary materials should be added to the main text. For example the mean recovery of elements +/- SD and uncertainty of results obtained for red and white wine samples.

The description of X-axis in Figure S1 is wrong.

Section 3. Figure numbers are mistaken. Please, correct them.

In my opinion the map of the country with wine regions marked will be advantageous for better understanding the results and similarities/differences obtained for wines from different regions. Moreover, the discussion on the sources of similarities/differences between REE fingerprints would be beneficial.

The example of proper identification of the unknown wine samples based on the obtained PCA results would also increase the significance of the paper.

Author Response

The paper presents interesting results on the using of rare earth elements fingerprint for the geographical discrimination of Spanish wines. The concept of using REE for discrimination of various wines is not novel, but the authors complemented it with analysis of other metals. Additionally, they applied a novel direct procedure for determination of elements in wines by ICP-MS.

Thank you for these comments. The manuscript has been entirely revised according to the referee´s comments.

Detailed remarks:

Introduction: lines 43-46: not all papers cited in this sentence describe using of REE for discrimination of wines. Please check and correct.

Thank you for the observation. References have been checked and the text has been accordingly modified.

Please describe also why REE can be used for geographical discrimination.

Thank you for the comment. A brief explanation has been included in the introduction section.

Novelty of the paper is not highlighted. Please, improve it.

Thank you for the comment. More relevance has been given to the novelty of the manuscript.

Different abbreviations are used in the paper for wine: PDO vs DPO. Please check and correct if necessary.

Thank you for the observation. The manuscript has been revised and all the abbreviations have been unified.

Section 2.1. The short description of results presented in Supplementary materials should be added to the main text. For example the mean recovery of elements +/- SD and uncertainty of results obtained for red and white wine samples.

The manuscript has been accordingly modified to introduce a short description of the results.

The description of X-axis in Figure S1 is wrong.

The Figure S1 has been modified.

Section 3. Figure numbers are mistaken. Please, correct them.

Thank you for this comment. All the figures numbers have been checked and modified.

In my opinion the map of the country with wine regions marked will be advantageous for better understanding the results and similarities/differences obtained for wines from different regions. Moreover, the discussion on the sources of similarities/differences between REE fingerprints would be beneficial.

Thank you for this suggestion. The PDOs map has been included as a supplementary figure.

The example of proper identification of the unknown wine samples based on the obtained PCA results would also increase the significance of the paper.

Thank you for your comment. We fully agree with this comment. When we carried out the wine analysis, there were several samples with an unknown PDO. We got access to this information later, and we introduced those data in the radar chart. We decided not to include a section on the manuscript due to a systematic application of the radar charts was not carried out. We applied this procedure for a limited number of samples. The truth is in the future we would like to complete this study and we will take this recommendation into account.

Reviewer 2 Report

In the manuscript ”Prospect on rare earth elements and metals fingerprint for the geographical discrimination of commercial Spanish wines” the authors address wine geographical origin problematic with the determination of rare earth elements (REEs) content by high-temperature torch integrated sample introduction system (hTISIS) combined with inductively coupled plasma mass spectrometry (ICP-MS). Wine authenticity has gained importance with the number of frauds increasing considerably. Determining wine's geographical origin as well as the development of suitable tools for that purpose is crucial. Therefore, I consider the theme proposed by the authors with relativity importance. The study is well-designed and planned and the results and discussion are well presented and discussed.

I recommend the publication of the manuscript with minor indicated amendments (presented below) in Molecules.

Specific comments:

  • English needs to be improved (inappropriate use of words in sentences changes their meaning);
  • Figures numbering should be in the correct order (confirm correspondence with the text);
  • I suggest inserting a table with the name of the elements or do it in the caption of each figure (caption should describe the figure properly without the need to read the text);
  • I also recommend a map with the location of the various wine regions analyzed that would be an asset to the article.

Author Response

In the manuscript ”Prospect on rare earth elements and metals fingerprint for the geographical discrimination of commercial Spanish wines” the authors address wine geographical origin problematic with the determination of rare earth elements (REEs) content by high-temperature torch integrated sample introduction system (hTISIS) combined with inductively coupled plasma mass spectrometry (ICP-MS). Wine authenticity has gained importance with the number of frauds increasing considerably. Determining wine's geographical origin as well as the development of suitable tools for that purpose is crucial. Therefore, I consider the theme proposed by the authors with relativity importance. The study is well-designed and planned and the results and discussion are well presented and discussed.

I recommend the publication of the manuscript with minor indicated amendments (presented below) in Molecules.

Thank you for these comments. The manuscript has been entirely revised according to the referee´s comments.

Specific comments:

English needs to be improved (inappropriate use of words in sentences changes their meaning)

Thank you for this comment. The manuscript has been entirely revised.

Figures numbering should be in the correct order (confirm correspondence with the text)

Thank you for this observation. The figure numbers have been modified and the manuscript has been accordingly modified.

I suggest inserting a table with the name of the elements or do it in the caption of each figure (caption should describe the figure properly without the need to read the text)

Thank you for the comment. The figures captions have been modified. Moreover, the elements analysed are listed in the Table S15.

I also recommend a map with the location of the various wine regions analyzed that would be an asset to the article.

Thank you for this suggestion. The PDOs map has been included as a supplementary figure.